# Assessing the Influence of Financial Inclusion on Environmental Degradation in the ASEAN Region through the Panel PMG-ARDL Approach

Seemab Ahmad [1], Dilawar Khan [1,*] and Róbert Magda [2,3]

1    Department of Economics, Kohat University of Science and Technology, Kohat 26000, Pakistan;
     seemiguli98@gmail.com
2    Institute of Agricultural and Food Economics, Hungarian University of Agriculture and Life Sciences,
     2100 Godollo, Hungary; magda.robert@uni-mate.hu
3    Vanderbijlpark Campus, Northwest University, Vanderbijlpark 1900, South Africa
*    Correspondence: dilawar@kust.edu.pk

**Abstract:** The rise of financial inclusion in recent years has attracted the attention of environmental economists to assess its role in environmental degradation. Therefore, this study was carried out with the aim of exploring the impact of financial inclusion on environmental degradation in the ASEAN region using balanced panel data for the period 2000–2019. First, panel unit root tests were employed to examine each data series for stationarity. Findings of the panel unit root tests depicted that all data series are stationary at the first difference. Second, Westerlund and Edgerton's error correction panel cointegration test was employed to handle heterogeneity and cross-sectional dependence. Third, the PMG-ARDL approach was used to explore the long- and short-term effects of financial inclusion on environmental degradation. Findings of the PMG-ARDL found that financial inclusion, energy use, economic growth and urbanization are causing environmental degradation in the ASEAN region. Furthermore, the financial inclusion coefficient is 0.15, which is statistically significant at 5%. In the short run, a 1% increase in financial inclusion results in a 0.15% increase in environmental degradation, ceteris paribus. In the long run, financial inclusion and $CO_2$ have a positive association that is statistically significant at 5% and has a coefficient value of 0.42. This implies that a 1% increase in financial inclusion results in a 0.42% increase in environmental degradation in the long run. Finally, this study recommends that financial inclusion must be incorporated into climate change adaptation efforts at the local, national and regional levels to address the side effects of increased $CO_2$ emissions.

**Keywords:** financial inclusion; environmental degradation; PMG-ARDL approach; ASEAN

## 1. Introduction

Climate change and global warming are challenges for the world today [1–3]. An anthropogenic climate has disastrous consequences for ecosystems, health and environmental sustainability that have drawn international concern. Human-induced greenhouse gas (GHG) emissions due to heavy dependence on fossil fuels and industrial discharges are the main causes of increasing global warming [4,5]. Carbon dioxide, which accounts for 70% of greenhouse gases, is commonly considered the main driver of environmental degradation among greenhouse gases (future GHGs) [6]. It is also widely expected that by 2035, atmospheric GHG concentrations will have doubled from pre-industrial levels. As a result, the rate of global warming could exceed the 2 °C threshold, and the world could face catastrophic consequences of climate change that affect all aspects of life, such as rapid sea-level rise, species extinction, severe droughts, water pollution and health complications [7,8]. At a global level, coal is the dominant fuel for power generation, and its share reached 36.4% in 2019. The energy sector is an important contributor to the growth



of global carbon emissions, given that fossil energy has serious negative environmental externalities [9,10].

Carbon emissions have been increased by a steady increase in world economic growth and humanitarian activities, highlighting the need for global $CO_2$ balance goals [11]. Consequently, the risk of ecological decline is growing daily across the world [12]. Protecting the environment has now become the biggest concern for all economies, with the fastest-growing challenge being the rapid increase in gas emissions [1,8,13–15]. GHG emissions, particularly $CO_2$ emissions, are the main drivers of global warming and climate change [16,17]. The term "Asian Century" has been coined since the 2008 Global Financial Crisis, and confirmation of this can be demonstrated by the fact that emerging Asian countries have been stimulating global economic growth. The ASEAN economic community, on the other hand, is likely to grow in the years ahead. Founded on the McKinsey Global Institute (MGI) report, Asian nations are among the best performing developing countries in the world with a direct long-term future [18]. While the socio-economic progress of the ASEAN region is remarkable, it is vital to recognize the region's ecological and environmental consequences.

With the rapid growth of financial inclusion in recent years, economists have been debating the effectiveness of financial inclusion in reducing/increasing environmental damage because it contributes to the expansion of the financial sector and provides sophisticated tools to financial intermediaries to promote the inclusive and sustainable growth of the economy. Financial inclusion is seen as an important part of economic development [19–21]. According to the World Bank [22], financial inclusion implies that all individuals and businesses have access to a wide range of simple-to-use, long-term and sustainable financial goods and services, such as transactions, payments, savings, credit and insurance. It can also allude to the ease with which individuals and companies can obtain financial assets. However, financial development is the process of reducing the costs of obtaining information, executing contracts and carrying out transactions, which has resulted in the creation of financial contracts, markets and brokers [23]. Globally, financing facilities are on the rise. Based on the 2018 Global Financial Inclusion Index report, two billion people have opened a bank account since 2011, with 515 million doing the same in 2014. Between 2014 and 2017, the global percentage of people who have a bank account or use a banking service online money increased from 62 to 69%. In emerging countries, the percentage rose from 54 to 63% [22].

Financial inclusion has the potential to reduce $CO_2$ emissions in both positive and negative ways. On the one hand, financial inclusion makes it simpler for companies and individuals to obtain low-cost financing programs that allow them to invest in green technologies. In this sense, inclusive financial systems can be extremely beneficial in reducing pollution by financing green technologies [24]. Financial involvement is especially important in disadvantaged communities because farmers are deprived of the financial resources to invest in carbon-free technologies. Farmers, for example, can buy an expensive solar microgrid that is more beneficial to the environment than coal-burning facilities if funding is available. Financial constraints (such as insufficient funds and government support) have been identified as major obstacles to adopting sustainable energy [25]. A financial system very well reduces borrowing rates, allowing companies to access the cash needed to increase production, resulting in increased carbon emissions [26]. Most importantly, the rise of social media and the internet has resulted in greater financial inclusion and access to electronic financial services [27,28]. According to 2017 Gallup World Poll statistics, 93% of adults in high-income countries own a smartphone, while 79% of those in low-income countries do [29]. According to the Global Financial Inclusion Index database, technological advances are critical to achieving the international fund's goal of a global financial approach by 2020 [22]. In addition to the repercussions on economic expansion, financial inclusion can influence environmental excellence. Improved financial inclusion, for example, encourages increased capital creation, which drives economic activity and, as a result, requires more energy. As a result, excessive energy use can have devastating

environmental implications [23]. Individual consumers may be able to buy more energy-consuming commodities such as electronic illustration devices [30], convertibles and other things because of the financial success global economy, increasing $CO_2$ emissions. Capital markets are seen as crucial markers of economic progress. Individual and institutional investors are attracted to the stock market due to its sustained performance, which drives production and consumption, resulting in increased $CO_2$ emissions due to the rampant use of fossil hydrocarbons [26]. According to this theory, ecological damage increases in tandem with financial deepening. This theory is supported by several experiential studies, which are [31,32]. Rising $CO_2$ emissions are due to increased manufacturing and industrial activity, as well as increased access to financial services, which cause global warming. Explosive growth can result in energy shortages, which can be a source of carbon emissions. However, expanding financial inclusion allows customers to purchase energy-intensive items such as cars, refrigerators and air conditioners, which pose a substantial environmental risk due to increased carbon emissions. Economic activities are facilitated by the integration of financial systems, which increases energy use and, as a result, carbon emission levels in the case of non-renewable energy) [33,34]. In a nutshell, economic development driven by financial inclusion has the potential to increase GHG emissions and degrade the environment. Concerns have arisen that the rapid development of these nations, which has been underpinned by rising rates of energy use, is being reflected in climate change. Fossil fuel ignition continues to contribute to global anthropogenic emissions through power plants, transportation, equipment and homes [35].

In the 21st century, the global economy is crucial. Developed and emerging countries are rapidly globalizing their economies as the financial sector expands. As a result, globalization can have an impact on economic growth as well as environmental implications. Financial inclusion accelerates and exacerbates this process by providing easy access to cash, bank loans and a variety of other financial schemes and investment instruments. Economic growth and progress require the integration of nations and regions; financial inclusion accelerates and enhances this process by providing easy access to cash, bank loans and a variety of other financial schemes and investment instruments.

The core objective of this study is to investigate the effect of financial inclusion on environmental degradation in ASEAN nations. ASEAN is an Association of Southeast Asian Nations. It is a political and economic union of 10 Southeast Asian member countries that promotes intergovernmental cooperation and facilitates political, economic, educational, military, security and sociocultural integration among its member countries [9]. ASEAN includes Brunei, Cambodia, Indonesia, Myanmar, Malaysia, Philippines, Singapore, Thailand and Vietnam. Laos was excluded from the study due to the unavailability of some necessary data. Second, the study used a financial inclusion index composed through principal component analysis. In this context, ASEAN economies may face a serious shift from environmental humiliation in the coming decades. Therefore, this region needs serious attention to suggest some effective and efficient policies regarding the environment. Unfortunately, much less literature was found on environmental challenges in the ASEAN region. The results of the study will provide policymakers in the ASEAN region with a clear and practical roadmap for implementing appropriate economic policies in the areas of financial inclusion and environmental degradation.

The contribution of this study is threefold: first, in view of the previous literature on environmental degradation, most studies were carried out to explore the factors that affect environmental degradation, ignoring the financial inclusion sector [36–39]. Second, the article's findings explore the relevance of financial inclusion for individuals as well as how to leverage financial inclusion to combat environmental degradation in the ASEAN region. Most studies used two or three financial inclusion components as a proxy for financial inclusion. This study developed a large financial inclusion index using the principal component analysis (PCA) approach to explore the impact of financial inclusion on environmental degradation in the ASEAN region. Third, this study is also a pioneering study exploring the impact of financial inclusion on environmental degradation in the ASEAN

region. In addition, this study also explores the impact of energy use, economic growth and urbanization on environmental degradation in the ASEAN region. Consequently, this study is also an updated addition to the existing stock of literature.

The rest of this article is organized as follows: Section 2 presents the literature review, and the methodology of the study was discussed in Section 3. Results and discussion are elaborated in Section 4. Conclusions and policy implications are deduced in Section 5. This section was followed by references.

## 2. Literature Review

Although several studies have focused on the economic impacts of financial inclusion, the connection between financial inclusion and $CO_2$ emissions has received little attention. Le et al. [24], for example, analyzed the effects of financial involvement, urban sprawl, foreign direct investment, energy use and industrialization on a panel of 31 nations. That study examined annual data from 2004 to 2014 and found that increased financial inclusion is associated with higher $CO_2$ emissions. Renzhi and Baek [40] found the influence of financial inclusion on $CO_2$ emission levels for 103 economies. The study found that financial inclusion decreases $CO_2$ emissions using the GMM technique and annual data from 2004 to 2014. It was found that financial inclusion can be a useful tool to reduce the negative consequences of economic development by promoting environmental awareness. Zaidi and Hassan [23] assessed the common dynamic consequences linked to economic collaboration and progress using annual data from 2004 to 2014. Access to finance, according to experts, lowers the level of carbon emissions in the long and short term.

Xing et al. [41] studied the relationship between financial development and $CO_2$ emissions in China from 2000 to 2013, concluding that: current financial development considerably increases $CO_2$ emissions, which vary by area; and (ii) financial inclusion follows an inverted U-curve. According to Zhang [42], financial deepening is one of the main contributors to the increase in China's $CO_2$ emissions. Shahbaz and Shahzad [43] investigated the influence of financial involvement on $CO_2$ emissions in Pakistan from 1985 to 2014 and found that as financial facilities improve, so does carbon intensity. Furthermore, Boutabba [44] revealed that the expansion of the financial sector had a favorable influence on $CO_2$ emissions in India for the years 1971–2008 and 1970–2012. Furthermore, according to Al-Mulali and Ozturk [31], capital accumulation had a favorable influence on $CO_2$ emissions in 23 European nations between 1990 and 2013. Fang and Yu [45] investigated the driving mechanism and decoupling effect of PM2.5 emissions from China's industrial sector growth. PM2.5 emissions are investigated by a refined Laspeyres index method. Empirical results illustrate that population distribution, the effects of coal consumption and industrial development are the main reasons for the increase in PM2.5 emissions that consequently contribute to environmental degradation. This leads to a serious public health impact in relation to infant mortality, depression, traffic accidents and increased stress hormones. Similarly, another work by Yu and Fang [10] employed the Generalized Divisia Index Method (GDIM), which integrates various influencing factors and provides a complete understanding of the PM2.5 pollution influence mechanism. The study explored the relationship between PM2.5 emissions and economic growth, and the contributions of technical and non-technical factors to the decoupling indicator are quantified. The study results showed that the technical effect plays an important role in promoting the decoupling between PM2.5 emissions and economic growth, but its contribution shows a downward trend over time. On the other hand, the non-technical effect hampers the decoupling process, and its contribution is decreasing from 2000 to 2014, which helps to improve air quality in China.

Acheampong [46] reported inconsistent results across 46 sub-Saharan African nations, demonstrating a positive influence of financial deepening on $CO_2$ emissions. The results of Salahuddin and Alam [47] are also ambiguous, as the long-term effect depends on the estimation approach. Finally, in the United Arab Emirates, Charfeddine and Ben Khediri [48] identified an inverted U-shaped link. In a dynamic panel framework, Saidi

and Mbarek [49] used time-series data from 19 rising countries from 1990 to 2013 to examine the influence of financial expansion on $CO_2$ emissions. Real data shows that when the financial system expands, $CO_2$ emissions fall, which means that financial expansion is environmentally friendly. Dogan and Seker [35] employed dynamic ordinary least squares (DOLS) and fully modified ordinary least squares to analyze the impact of financial development on $CO_2$ emissions using data from the top 23 renewable energy-consuming nations from 1985 to 2011. Findings showed that long-term financial inclusion and $CO_2$ emissions are linked, and access to finance lowers GHG emissions. From 1960 to 2013, Cetin and Ecevit [50] investigated the long-term and short-term causal effects of financial development on $CO_2$ emissions in Turkey. The approach they used was the dashboard configuration. Statistics have shown that the growth of the financial system has a beneficial influence on $CO_2$ emissions and that there is a unidirectional causal relationship between financial development and $CO_2$ emissions. Ali et al. [51] employed the ARDL approach to study the association between Nigerian financial inclusion and $CO_2$ emissions from 1971 to 2010. They found a favorable long-term and short-term relationship between financial development and $CO_2$ emissions. Statistics show that the growth of the financial system is linked to an increase in ecological consequences.

Some gaps in the literature may be discovered after reviewing previous content. Most of the previous literature has focused on financial development, with little research looking at the connection between financial inclusion and carbon emissions. No studies have explored the effect of financial inclusion on $CO_2$ emissions in the ASEAN region. Furthermore, although the previous study focused on the use of renewable energy, only a few studies looked at environmental degradation.

## 3. Materials and Methods

The main objective of this research was to explore how financial inclusion affects $CO_2$ emissions in ASEAN countries. Laos was excluded from the study due to a lack of data on numerous variables. The variables used are listed in Table 1. To capture the influence of financial inclusion as the main explanatory variable, the study used $CO_2$ emissions as a proxy for environmental degradation, in addition to several indicators such as automated teller machines (ATMs), bank branches, bank deposits and life insurance premiums. Control variables in the present study include energy use, urbanization and GDP. The World Development Indicators provide data on carbon emissions, economic development (GDP) and energy use [22]. Financial inclusion data is taken from the World Bank's Global Financial Development Database (GFDD). Data for urbanization were taken from the United Nations website (https://population.un.org/wup/DataQuery/, accessed on June 23, 2021). The study period covers annual data from 2000 to 2019.

**Table 1.** Description of the variables used in the study.

| Abbreviations | Variables | Unit | Data Source |
|---|---|---|---|
| $CO_2$ | Carbon emissions | Metric tons per capita | World Bank (2021) |
| | This index is comprised of the following components: | | |
| | Automated teller machine (ATM) | Index | World Bank (2021) |
| FI | | per 100,000 adults | World Bank (2021) |
| | 1.  Bank branches | per 100,000 adults | World Bank (2021) |
| | 2.  Bank deposits | GDP (%) | World Bank (2021) |
| | 3.  Life insurance premium | Volume to GDP (%) | |
| GDP | Gross Domestic Product | (Million measured at constant 2015 US$) | World Bank (2021) |
| EU | Energy use | (kg of oil equivalent per capita) | World Bank (2021) |
| URB | Urbanization | In thousands | United Nations (2021) |

Source: World Bank (2021).

The key dependent variable, environmental degradation, is represented by $CO_2$ emissions. Unless $CO_2$ emissions decrease, climate change will continue to have irreversible negative repercussions for Earth's species. Global climate change is mainly caused by greenhouse gas emissions. $CO_2$, which is primarily a by-product of energy generation and use, is responsible for most of the greenhouse gases linked to global warming. The burning of fossil fuels and the production of cementitious materials are the main sources of man-made carbon emissions [24,52]. Financial inclusion is the main explanatory variable. We generated a composite Financial Inclusion Index that integrates five financial inclusion parameters to build this index. Financially incorporated individuals and businesses have access to suitable and acceptable monetary goods and services that meet their needs, such as purchases, transactions, deposits, credits and insurance, all of which are offered on a sustainable basis. Greater financial access helps and promotes industrial production and activity, potentially increasing the level of carbon emissions into the atmosphere, which is a major contributor to climate change. In addition, due to the increase in financial inclusion, families tend to use more products such as cars, TVs, computers, ovens, freezers, air conditioners and similar ones, leading to an increase in total energy consumption and, consequently, emissions of $CO_2$. Finally, increased financial inclusion drives overall economic activity, resulting in increased $CO_2$ emissions [53–55].

Gross domestic product (GDP) is a monetary indicator of the total monetary value of all final goods and services produced during a given period. Most carbon dioxide emissions are produced using fossil fuels such as coal, which is the main source of energy for the automotive industry and is directly linked to economic growth and development. For the execution of related strategies, it is critical to understand the direction of the causal link between economic development, energy use and $CO_2$ emissions [56,57]. The amount of energy used to perform a task, produce goods, or simply live in a building is called energy usage. Energy, along with land, labor, money and entrepreneurship, is an essential resource input component in the manufacturing process. As a result, energy use impacts production; hence carbon emissions grow. As a result, any policy aimed at reducing emissions must include measures to promote the production and use of renewable energy [58]. Economic development and energy use, in this view, indicate the amount of $CO_2$ emissions, which is the main ecological disaster [28,59–61]. The process of concentrating large numbers of people in relatively limited regions to develop cities is known as urbanization. Positively, urbanization is linked to ecological damage [62].

Table 1 shows the variables, units and their sources used in this study:

Table 2 summarizes the descriptive statistics of the data series used in the study, such as the mean, median and standard deviation. The mean value of the dataset is the center value of the dataset. Standard deviation is a metric that measures how far the data deviates from the mean. A smaller standard deviation number implies that the dates are concentrated around the mean value. Furthermore, it ensures that the statistical results are reliable and efficient [63]. Brunei has the highest average $CO_2$ emissions (16.14 metric tons per capita), while Myanmar has the lowest average (0.28 metric tons per capita). The financial inclusion index has the highest average value (4.57) in Singapore, while it has the lowest average value in Myanmar (0.37). Brunei has the highest energy use (8067 kg of oil equivalent per capita), but Singapore's average value is the lowest (71.73 kg of oil equivalent per capita). Indonesia has the highest GDP (670,318 million measured at constant 2015US$), while Brunei has the lowest GDP (12,763.94 million measured at constant 2015US$). Cambodia has the lowest average urbanization value (299 thousand), while the Philippines has the highest average urbanization value (4283.5 thousand).

**Table 2.** Descriptive statistics of the variables.

| Countries | Variables | Mean | Median | Std. Dev. |
|---|---|---|---|---|
| Brunei | $CO_2$ | 16.14 | 16.64 | 2.272 |
| | FI | 3.57 | 3.66 | 0.407 |
| | EU | 8067.52 | 8511.59 | 1155.81 |
| | GDP | 12,763.94 | 12,838.67 | 565.59 |
| | URB | 290.05 | 288.05 | 31.99 |
| Cambodia | $CO_2$ | 0.36 | 0.33 | 0.19 |
| | FI | 0.83 | 0.64 | 0.71 |
| | EU | 136.71 | 131.22 | 27.44 |
| | GDP | 13,345.4 | 12,391.3 | 5427.6 |
| | URB | 2958.65 | 2856.84 | 522.54 |
| Indonesia | $CO_2$ | 1.73 | 1.68 | 0.25 |
| | FI | 1.84 | 1.34 | 0.79 |
| | EU | 106.15 | 104.7 | 14.12 |
| | GDP | 670,318 | 638,563.5 | 208,498.2 |
| | URB | 119,393 | 119,325 | 19,527.2 |
| Malaysia | $CO_2$ | 6.8 | 7.04 | 0.79 |
| | FI | 3.8 | 3.72 | 0.23 |
| | EU | 131.84 | 133.3 | 6.14 |
| | GDP | 239,190.8 | 226,277.7 | 68,136.07 |
| | URB | 19,648.2 | 19,639.1 | 3315.5 |
| Myanmar | $CO_2$ | 0.28 | 0.22 | 0.15 |
| | FI | 0.31 | 0.18 | 0.33 |
| | EU | 128.77 | 95.98 | 53.81 |
| | GDP | 41,426.89 | 40,598.24 | 19,772.39 |
| | URB | 14,488.5 | 14,389.4 | 1293.45 |
| Philippines | $CO_2$ | 0.98 | 0.9 | 0.17 |
| | FI | 1.75 | 1.55 | 0.32 |
| | EU | 81.29 | 73.17 | 14.91 |
| | GDP | 240,040.9 | 220,810.8 | 78,340.55 |
| | URB | 42,834.5 | 42,134.2 | 4584.4 |
| Singapore | $CO_2$ | 8.69 | 8.39 | 0.77 |
| | FI | 4.57 | 4.52 | 0.23 |
| | EU | 71.73 | 63.5 | 17.31 |
| | GDP | 238,928.3 | 231,408.5 | 72,621.77 |
| | URB | 4959.6 | 5020 | 620.98 |
| Thailand | $CO_2$ | 3.4 | 3.45 | 0.38 |
| | FI | 4.14 | 4.05 | 1.1 |
| | EU | 171.21 | 121.05 | 223.96 |
| | GDP | 339,837.2 | 335,966.6 | 73,101.38 |
| | URB | 28,212 | 28,964 | 4908.09 |
| Vietnam | $CO_2$ | 1.56 | 1.57 | 0.65 |
| | FI | 0.74 | 0.69 | 0.28 |
| | EU | 123.62 | 123.49 | 3.99 |
| | GDP | 148,062.4 | 140,618.1 | 52,959.23 |
| | URB | 26,956 | 26,486 | 5036.67 |

Source: Authors' calculations.

Before using the Pooled Mean Group-Autoregressive Distributed Lag (PMG-ARDL) approach, this study investigated cross-sectional dependence (CD). Some non-stationarity tests from the first-, second-and third-generation tests were applied to solve CD problems. The problem of cross-sectional dependence must be addressed, as it has led to erroneous results, size distortion, stationarity bias, and cointegration, among other issues [15,64,65]. We employed the [66] CD test to assess whether there is any cross-section dependence issue. The panel data unit root approach was used once the cross-section dependency values had

been determined. As fundamental first-generation techniques are insufficient to deal with cross-dependence, second-generation approaches were used. This study is based on [66] methodology to deal with cross-sectional dependence, as the existence of CD contradicts the conclusions of the first-generation test [67,68].

$$CD = \sqrt{2T/N(N-1)} \left( \sum_{i=1}^{N-1} \sum_{j=i+1}^{N} \hat{p}\, ij \right) \qquad (1)$$

This study applied three unit-root tests: Pesaran's cross-sectional augmented Dickey–Fuller (CADF), Breitung and Das, and cross-sectionally augmented Im, Pesaran and Shin (CIPS) tests. The $H_0$ argues that the variable is I(0), while the $H_a$ stated that variables are stationary at the first difference I(1).

$$CIPS = \frac{1}{N} \sum_{i=1}^{N} CADF\, i \qquad (2)$$

The alternative hypothesis, on the other hand, states that at least one unit must be connected. It is a second-generation cointegration test that gives us more accurate and meaningful information about the long-term connection between parameters. $H_0$ argues that there is no long-term relationship between the variables, while the alternative hypothesis clarifies that there is a long-term relationship. We accept alternative cointegration hypotheses if the prob value of the Z statistic is less than the critical level. In this article, we examine the impact of financial inclusion on environmental degradation in ASEAN countries.

We used the PMG-ARDL approach to explore the impact of financial inclusion on environmental degradation. The standard panel models such as pooled OLS, fixed-effects and random-effects models have some serious shortcomings. For example, pooled OLS is a highly restrictive approach by imposing common intercept and slope coefficients for all cross-sections and, therefore, disregards individual heterogeneity. On the other hand, the fixed-effects model assumes that the estimator has common slopes and variance but country-specific intercepts [69]. Furthermore, the parameter estimates produced by the fixed effects model are biased when some regressors are endogenous and correlated with the error terms [70]. In contrast to the fixed-effects model, the random-effects model is relatively less problematic in terms of degrees of freedom by assuming common intercepts. However, the random-effects model has another shortcoming in that it considers the model to be time-invariant [71]. In real life, this assumption is often invalid. Additionally, static panel estimators do not take advantage of the panel dimension of the data by decomposing between the short and long-run relationships [72]. The above discussion concludes that the standard panel models are unable to capture the dynamic nature of the data, which is a fundamental issue in the empirical literature. In addition, these estimators can only deal with the structural heterogeneity in the form of random or fixed effects but impose homogeneity in the models' slope coefficients across countries even when there may be substantial variations between them.

Keeping in view the above shortcomings of standard panel models, this study employed a dynamic panel heterogeneity approach developed by [73,74] to examine the short and long-run relationships between financial inclusion and environmental degradation. By creating indicators of (weighted) cross-sectional averages of regressors to control for the common factor and solve the contemporaneous correlation issue, this study used panel ARDL of the Common Correlated Effect Pooled Mean Group (CCEPMG). It is expected that CCEPMG will be consistent and efficient in this estimation under the null hypothesis of no heterogeneity in the long run. This study uses the data of nine countries in the ASEAN region over a period of 20 years (2000–2019), resulting in a total of 180 observations. Since the data consists of a panel of 9 countries for 20 years, where $N = 9$ is much less than $T = 20$, the GMM estimator is not appropriate for our analysis. However, when $T$ is larger than N (as in this case), the PMG-ARDL approach is appropriate and therefore is the preferred method for our analysis. Moreover, the use of two different estimators

such as the mean group (MG) and the pooled mean group (PMG), will explore (1) the long and short term effects of financial inclusion on environmental degradation in the ASEAN countries and also (2) to capture the speed of convergence towards its equilibrium. The number of time series is relatively large than the cross-section ($T > N$). For large $T$, [75] show that traditional panel techniques (fixed effect, instrumental variables, GMM estimators) can produce inconsistent and potentially misleading estimates of the average parameter values in the dynamic panel data model unless the slope coefficients are, in fact, identical. Therefore, the relationship between environmental degradation and financial inclusion was examined using the following model:

$$CO_{2i,t} = \beta_o + \beta_1 FI_{it} + \beta_2 EU_{it} + \beta_3 GDP_{it} + \beta_4 URB_{it} + e_{it} \tag{3}$$

where, $i = 1 \ldots N$ represents the countries and $t = 1, \ldots, t$ indicates the time period. $CO_2$ is a proxy used for environmental degradation; financial inclusion (*FI*) is an index composed of principal components analysis for financial inclusion; *EU* is used for energy use, and *GDP* is used for economic growth. Furthermore, $\beta_0$, $\beta_1$, $\beta_2$, $\beta_3$ and $\beta_4$ are the parameters to be estimated, $e_{it}$ is the residual term and must be normally distributed with zero mean and constant variance, i.e., $N(0, \delta^2)$.

The short-term and long-term impacts of financial inclusion on environmental degradation were examined using the PMG-ARDL model established by Pesaran, Shin et al. ([66]). The pooled mean group constrains long-run coefficient homogeneity across the countries, but short-run dynamics remain heterogeneous. The Mean Group (MG) does not impose restrictions on the coefficients, allowing them to change and be heterogeneous in the short and long term. Under long-term homogeneity, the PMG estimator, according to ([66], provides an improvement in the efficiency estimates in relation to the Mean Group estimator (MG). The empirical model used in this study is as under:

$$
\begin{aligned}
\Delta Ln CO_{2i,t} = \beta_o \quad & + \sum_{e=1}^{a-1} \beta_{1e} election\ criteria\ ture \Delta Ln CO_{2i,t-e} + \sum_{e=0}^{a-1} \beta_{1e} election\ criteria\ ture \Delta Ln FI_{i,t-e} \\
& + \sum_{f=0}^{b-1} \beta_{2f} \Delta Ln EU_{i,t-f} + \sum_{g=0}^{c-1} \beta_{3g} \Delta GDP_{i,t-g} + \sum_{h=0}^{d-1} \beta_{4h} \Delta Ln URB_{i,t-h} \\
& + \delta_1 Ln CO_{2i,t-1} + \delta_2 Ln FI_{i,t-1} + \delta_3 Ln EU_{i,t-1} + \delta_4 Ln GDP_{i,t-1} + \delta_5 Ln URB_{i,t-1} \\
& - \delta_{881IITive\ results(1)come} ECT_{i,t-1} + e_{it}
\end{aligned} \tag{4}
$$

where $\beta$ and $\delta$ are short- and long-term parameters, and $e_{it}$ represents residuals of country $i$ at time $t$ and $\Delta$ is a difference operator after rendering the data stationary. The term $ECT_{i,t-1}$ represents the error correction term; this term defines the model's dynamic stability. The model is regarded as dynamically stable if the ECT value is negative and significant, implying that the short-run instability in the model is automatically balanced in the long run. Finally, to verify the consistency, precision and constancy of the variables used in the PMG-ARDL technique, the study performed a variety of diagnostic statistics, i.e., the Hausman test, the normality test and the Ramsey RESET test.

## 4. Results and Discussions

The financial inclusion index was computed using the principal component analysis (PCA) approach from four components (number of ATMs, number of bank branches, bank deposits and life insurance premium) [40]. The units and scales of the financial depth metrics vary. In addition, certain variables show high volatility, while others show low variance. As a result, before combining these indicators to generate a composite index, they were transformed into normalized variables. The normalized variables are then subjected to principal component analysis [20]. The principal component analysis (PCA) is the most used indexing method in the literature for several reasons. It is a famous and standard approach that identifies hidden features and correlations, as well as removes extraneous data, minimizes data dimensionality and builds a composite indicator [76]. The Principal Component Analysis and Factor Analysis are two data reduction approaches

that are employed to confirm multidimensional datasets with little disturbance. The PCA does not demonstrate the existence of limited common factors causing variation in the data, whereas factor analysis could. Table 3 shows the results of the aggregation of the financial inclusion indices by the PCA method. The correlation matrix of the FI index components is presented in the first section of the table. The Eigen values, variance, percentage and commutative proportion of the four components of financial inclusion are shown in the second section. Figure 1 shows the trend of financial inclusion in ASEAN countries. Moreover, Figure 2 shows the trend of carbon emission measured in metric tons per capita in the ASEAN region.

**Table 3.** Results of principal component analysis.

| Variable | ATMs | Bank Branches | Bank Deposits | Life Insurance Premium |
|---|---|---|---|---|
| | | Correlation Matrix | | |
| AT | 1 | | | |
| BB | 0.69 | 1 | | |
| BD | 0.68 | 0.46 | 1 | |
| LIP | 0.55 | 0.19 | 0.76 | 1 |
| | | Component Analysis | | |
| Component | Eigenvalue | % Variance | Cumulative proportion % | First principal Component |
| 1 | 2.68 | 0.67 | 0.67 | (0.55) [AT] |
| 2 | 0.90 | 0.22 | 0.89 | (0.43) [BB] |
| 3 | 0.24 | 0.06 | 0.95 | (0.55) [BD] |
| 4 | 0.18 | 0.05 | 1 | (0.47) [LIP] |

Source: Authors' calculations.

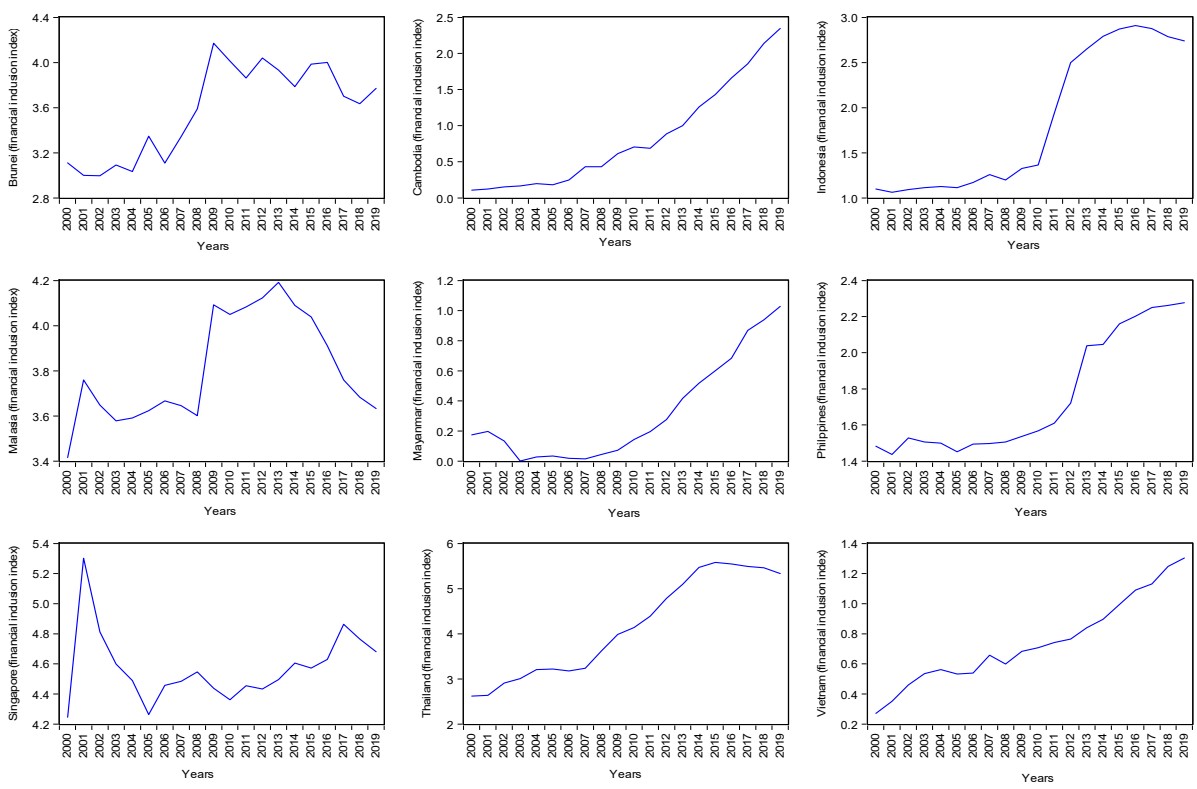

**Figure 1.** The trend of financial inclusion in ASEAN countries.

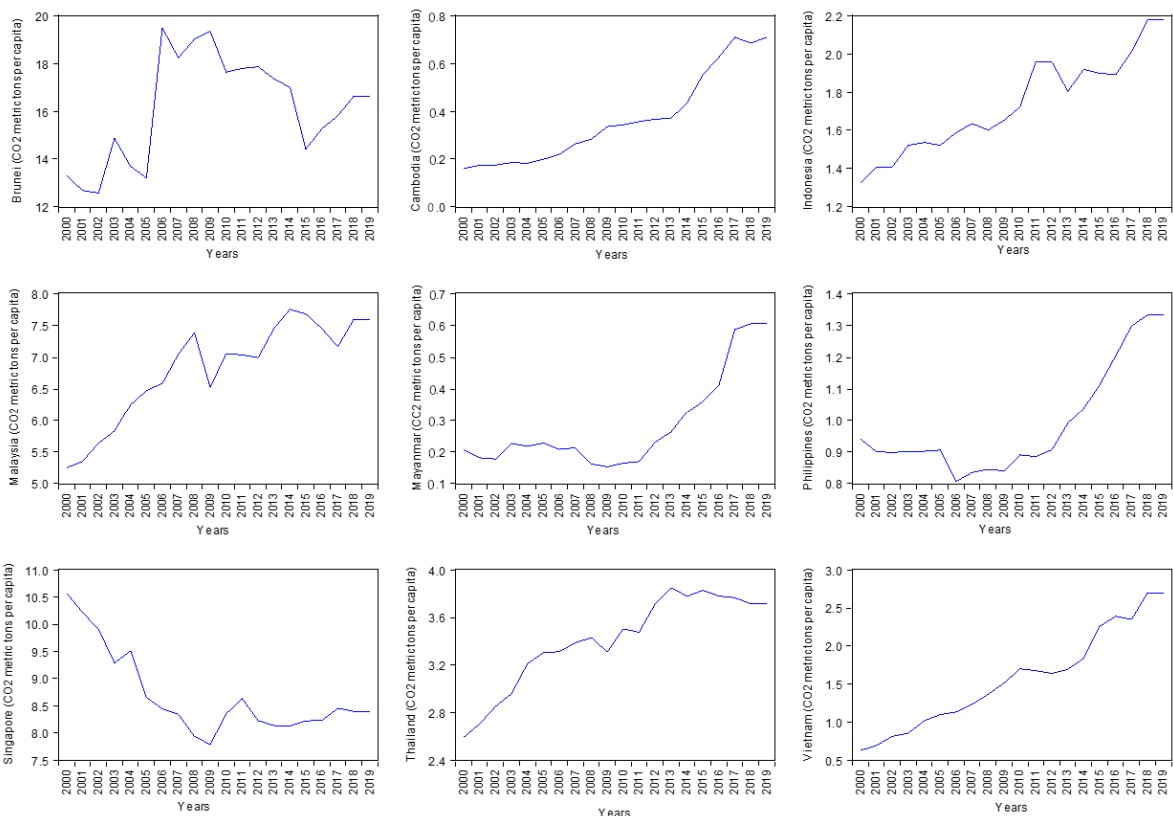

**Figure 2.** The trend of carbon emissions in ASEAN countries.

In this study, stationarity tests were used to investigate non-stationarity in each data series. Non-stationary data series can result in inaccurate information and unforeseen consequences because most economic data series follow a unit root process. To obtain unit root-free data, three second-generation panel unit root tests were used in the analysis: Pesaran CADF [66], Breitung and Das test [77] and CIPS test. All three second-generation panel stationarity tests demonstrated that all data series are stationary at first order I (1) at both intercept and intercept and trend. The results of the panel unit root tests are shown in Table 4.

**Table 4.** Second-generation panel unit root results.

| Series | Breitung and Das Test | | CIPS Test | | CADF Test | |
|---|---|---|---|---|---|---|
| | **Level** | **First Diff** | **Level** | **First Diff** | **Level** | **First Diff** |
| | Intercept | | | | | |
| $CO_2$ | −2.42 | −3.23 ** | −1.09 | −2.75 ** | −1.76 | −2.07 ** |
| FI | −1.86 | 3.33 ** | −2.06 | 3.11 ** | −1.70 | −2.83 ** |
| EU | −2.08 | −2.79 ** | −1.71 | −2.92 ** | −2.21 | −2.95 ** |
| GDP | −2.08 | −2.79 ** | −2.14 | −3.10 ** | −2.41 | −3.21 ** |
| URB | −2.26 | −2.97 ** | −2.27 | −3.71 ** | −2.91 | −4.17 ** |
| | Intercept and trend | | | | | |
| $CO_2$ | −1.18 | −4.15 ** | −2.13 | 3.48 ** | −0.93 | −2.45 ** |
| FI | −1.94 | 2.84 ** | −2.92 | 4.03 ** | −2.23 | 3.37 ** |
| EU | −2.23 | −3.37 ** | −2.37 | 3.68 ** | −1.86 | −3.29 ** |
| GDP | −1.03 | −2.93 ** | −1.86 | −3.85 ** | −2.05 | −3.41 ** |
| URB | −1.49 | −3.52 ** | −1.10 | −2.69 ** | −2.36 | −3.94 ** |

Source: Authors' calculations. ** shows significance at 5%.

The next step is to explore the cointegration between environmental degradation and financial inclusion after confirming that each series is stationarity in the first difference

I (1). The results of Westerlund's cointegration test are shown in Table 5. Westerlund's [65] cointegration approach is better compared to previous cointegration tests as it adds cross-sectional dependence. As a result, at the 1% and 5% significance levels, Group-t ($G_t$), group-a ($G_a$), Panel-t ($P_t$) and Panel-a ($P_a$) are the four components of the Westerlund second-generation error correction panel cointegration test reject the null hypothesis ($H_0$) of non-cointegration, confirming the presence of cointegration between environmental degradation and financial inclusion and other variable controls. Moreover, these criteria can be leveraged to further reduce $CO_2$ emission factors. Westerlund's cointegration test method demonstrates that environmental degradation and financial inclusion, energy use, GDP and urbanization have a co-integration link. The findings of the Westerlund cointegration assessment are shown in Table 5.

**Table 5.** Findings of Westerlund second-generation error correction panel cointegration test.

| Statistics | Values | Z-Values | Prob. |
| --- | --- | --- | --- |
| $G_t$ | −3.21 * | −5.97 | <0.01 |
| $G_a$ | −15.51 * | −8.49 | <0.01 |
| $P_t$ | −9.21 * | −5.12 | <0.01 |
| $P_a$ | −12.34 * | −10.01 | <0.01 |

Source: Authors' calculation; * 1% significance level.

Several diagnostic evaluations were applied to verify the characteristics of the parameters, including the Hausman test to verify if there was a significant difference between the PMG and MG estimators. According to the Hausman test, the null hypothesis ($H_0$) is that the pooled mean group (PMG) is preferred, while the H1 is that the mean group (MG) is preferred Hausman, [78]. The omitted variable bias test was used to examine whether the regressor and the omitted variable had a connection, and the normality assessment was employed to verify whether the residual error was normally distributed [29]. The Ramsey RESET (Regression Equation Specification Error Test) test was also used to ensure that the model was specified correctly [79]. We accept alternative hypothesis $H_1$ if the probability value of the F statistic is less than 5% (0.05), suggesting that the model is not the most adequate; we accept the null hypothesis $H_0$ if the prob-value of the F statistic is greater than 5% (>0.05), indicating that the model used in the study is the most adequate. Table 6 shows the results of the diagnostic tests.

**Table 6.** Results of diagnostic tests.

| Diagnostic Tests | Test Statistics | *p*-Values |
| --- | --- | --- |
| Hausman test | 1.41 | 0.12 |
| Normality test | 0.89 | 0.71 |
| Ramsey RESET test | 1.52 | 0.63 |

Source: Authors' calculations.

Finally, this study employed the PMG-ARDL approach to examine the long-term and short-term impacts of financial inclusion, energy use, gross domestic product (GDP) and urbanization on environmental degradation. The homogeneity of the long-term coefficient across countries is limited by the PMG, while the short-term dynamics remain varied. MG does not require short-term limits and allows parameters to change and be varied. Furthermore, when compared to the mean group estimates under long-term homogeneity, the Pool Mean Group estimator outperforms the MG estimator in terms of efficiency estimates [66]. The Hausman test was performed to select the appropriate one between the PMG and MG estimators Hausman, [78]. The variation of the estimated coefficients between the MG and the PMG is not statistically significant under the H0, the PMG being the appropriate one. The Hausman test, which is based on the Chi-square distribution, has an estimated value of 1.41, which is not statistically significant at 5%. We found that the PMG estimator is preferable compared to the MG estimator if the null hypothesis ($H_0$) is

accepted. The results of PMG-ARDL are shown in Table 7. Findings of the PMG-ARDL explored that financial inclusion (FI), energy use (EU), gross domestic product (GDP), and urbanization (URB) positively and significantly cause environmental degradation in the long run in the ASEAN region.

**Table 7.** Results of the PMG-ARDL approach.

| Variables | Parameters | t-Values | Std. Error | Prob. |
|:---:|:---:|:---:|:---:|:---:|
| LFI | 0.42 | 6.12 * | 0.07 | <0.01 |
| LEU | 0.67 | 7.75 * | 0.08 | <0.01 |
| LGDP | 1.08 | 9.01 * | 0.11 | <0.01 |
| LURB | 1.36 | 5.69 * | 0.24 | <0.01 |
| ΔLFI | 0.01 | 0.15 ** | 0.07 | <0.05 |
| ΔLEU | 0.28 | 1.04 | 0.26 | 0.29 |
| ΔLGDP | 0.75 | 1.60 ** | 0.47 | <0.05 |
| ΔLURB | 5.008 | 1.81 ** | 0.75 | <0.05 |
| Constant | −2.26 | −2.73 ** | 0.82 | <0.05 |
| ECT | −0.33 | −2.85 * | 0.11 | <0.01 |

Source: Author's calculations; * 1% significance and ** 5% significance level.

The financial inclusion coefficient is 0.15, and the prob value is 0.05, which is statistically significant at 5%. In the short term, a 1% increase in the financial inclusion results in a 0.15% increase in environmental degradation (carbon emissions), ceteris paribus. In the long run, financial inclusion (FI) and $CO_2$ have a positive association that is statistically significant at 5% and has a coefficient value of 0.42. Financial inclusion in the ASEAN region appears to have resulted in higher $CO_2$ emissions. Effective financial inclusion in ASEAN countries, according to the results, will increase environmental destruction, particularly in the long term. It argues that an affordable financial system has adverse long-term consequences for environmental quality because it supports the production and industrial activities through low-cost financing, which increases $CO_2$ emissions [80]. Through microcredit, individual consumers are also encouraged to adopt energy-efficient appliances that consume more energy [30].

The energy use coefficient is 0.28, with a prob value of 0.29, which is statistically insignificant at 5%. Other factors being held constant, a 1% increase in the value of the energy use coefficient leads to a 0.28% increase in environmental degradation in the short term. Energy use and $CO_2$ have a positive long-term relationship, which is significant at 5% and has a coefficient value of 0.64. Similar findings were also found by [81,82]. Asia's growing energy consumption is a result of the region's economic growth, urbanization and industrialization. The manufacturing, transportation, energy, heating and residential sectors have seen an increase in energy consumption because of Asia's rapid industrialization and urbanization. The use of these energy-intensive commodities jeopardizes the long-term stability of the environment. Energy consumption and GDP have a positive and considerable influence on environmental deterioration in the ASEAN regions in the short and long term. Boutabba [44] also found the same relationship between energy use, economic growth and $CO_2$ emissions, which we validated. They found that while the economic boom and the use of non-renewable energy increase carbon emissions, the use of renewable energy decreases gaseous emissions. In addition, gross domestic product (GDP) has a coefficient value of 0.75, and the *p*-value is 0.05, which is statistically significant at 5%. If all other factors are held constant, a 1% increase in the national income led to a 0.75% increase in carbon emissions in the short term. In the long term, national income and $CO_2$ have a substantial positive association, with a coefficient value of 1.08. [83] concluded that economic expansion, energy use and urbanization are the most important accelerators of carbon emissions. The influence of energy use, urbanization and economic expansion on carbon emissions has also been addressed in recent studies by Abbasi et al. [84]; Adedoyin and Zakari [85]; Parker and Bhatti [86]; Fazal et al. [87]. Moreover, the urbanization coefficient is 1.36, and the *p*-value is 0.05, which is statistically significant at 5%. If all other factors

remain constant in the short term, a 1% increase in the value of urbanization translates into a 1.36% increase in the present value of carbon dioxide emissions. With a coefficient value of 5.008, there is a direct and significant link between urbanization and environmental degradation in the long run. According to previous studies, energy use, urbanization and industrialization are expected to have a positive influence on $CO_2$-emissions [88,89]. Furthermore, the error correction term (ECT) indicates that the model is dynamically stable, which means that short-term disequilibrium is eliminated over time. For the persistence of dynamic stability, the ECT coefficient must be negative and significant. In the results, the coefficient of the error correction term is $-0.33$ absolutely corresponds to the first assumption of a significant negative ECT term of 1%. The estimated PMG-ARDL approach has an annual adjustment effect of 33% and is dynamically stable. This illustrates how the short-term mismatch will progressively correct itself over the next 3.03 years.

## 5. Conclusions and Policy Implications

This study explored that the investigated factors such as financial inclusion, energy use, gross domestic product and urbanization seem to have driven the increase in $CO_2$ emissions in ASEAN countries during the period 2000–2019. Financial inclusion has substantial environmental costs, but that does not mean it should be reduced. Rather, governments must use more technical ways to improve financial inclusion and access to capital. China, for example, is a major player in green finance with significant policy implications. The "Green Credit Guidelines", unveiled on 24 February 2012, are hailed as a pivotal moment in China's green monetary policies, demanding that Chinese financial systems support the growth of a green economy, reduce carbon levels in the economy, continue to improve their green credit financial services criteria and expand green credit mortgage capacity [90]. In all country administrations, financial inclusion programs must be linked to environmental protection. Governments must extend their approach and inclusion of climate subsidies to help poor and challenged segments of society tackle rising levels of $CO_2$ emissions. Micro, small and medium-sized enterprises, as well as consumers, must have sufficient access to financial products and services to engage in small-scale, local $CO_2$ reduction and adaptation programs.

## 6. Limitations and Prospects of the Study

This study used data from the period 2000 to 2019 due to data availability and excluded Laos due to a lack of data on numerous variables. The extended sample to date and covering the entire ASEAN region will provide a better understanding of the issue. This study also did not take into account the possible influence of bilateral trade/cooperation between ASEAN countries. Furthermore, the study can be extended to a global scale, comparing the role of financial inclusion of different regions in environmental degradation compared between SAARC, Europe and ASEAN regions. In addition, some factors such as industrial waste, deforestation and many others need to be considered.

**Author Contributions:** Conceptualization, S.A. and D.K.; methodology, D.K. and R.M.; software, S.A. and D.K.; validation, D.K., S.A. and R.M.; formal analysis, S.A. and D.K.; investigation, D.K. and R.M.; resources, R.M.; data curation, S.A., D.K. and R.M.; writing—original draft preparation, S.A., D.K. and R.M.; writing—review and editing, S.A. and D.K.; visualization, D.K. and R.M.; supervision, D.K. and R.M.; project administration, S.A. and R.M.; funding acquisition, R.M. All authors have read and agreed to the published version of the manuscript.

**Funding:** This research received no external funding.

**Institutional Review Board Statement:** Not applicable.

**Informed Consent Statement:** Not applicable.

**Data Availability Statement:** Data is openly accessed and freely available to everyone.

**Conflicts of Interest:** The authors declare no conflict of interest.

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
