# Peer review of "Assessing the Influence of Financial Inclusion on Environmental Degradation in the ASEAN Region through the Panel PMG-ARDL Approach"

_sustainability, doi:10.3390/su14127058_

Round 1

Reviewer 1 Report

The study reveals the effect of financial inclusion on environmental degradation, in the case of South Asia during 2000-2019. However, the paper must address some problems before it could be published

  1. The authors shall provide the novel contribution and significance of the study in the Introduction, not just merely highlighting ASEAN's condition. Otherwise, this would be difficult to highlight the research gaps from an international perspective and link them to the objectives of the study.
  2. It is suggested to give the definition or meaning of financial inclusion. ALSO, please explain the difference between financial development and financial inclusion, so readers can better understand the contribution of this study.  
  3. Why you should use ARDL-PMC estimation not other techniques to study the effects? And what are their advantages over traditional econometric methods? More explanation is necessary.
  4. I am confused about the research sample for this article, how many countries did you use, all Asian countries or South Asian countries, and which countries?
  5. What is the meaning of ATM?
  6. Move Table 3 to Section 3.
  7. Lack of theoretical basis in Eq. (3). According to the STIRPAT model, environmental degradation/carbon emissions will be affected by three factors: population, affluence, and technology. You can find evidence in a lot of literature. You should kindly explain why you choose these variables?
  8. To broaden the view of readers and display the innovation in this paper, more recently published pollution emissions & environmental policy-related studies should be added and discussed, e.g., the following literature:

https://doi.org/10.1016/j.ecolind.2021.107795

https://doi.org/10.1016/j.enpol.2020.112017

https://doi.org/10.1016/j.energy.2021.121295

https://doi.org/10.1016/j.enpol.2021.112668 

The authors need to provide a structured synthesis of them in the literature review or other related parts.

  1. Check the Table numbers, there are two Table 6. In addition, Tables 5 and 6 should be three-line tables. What is the significance level in Tables 5 and 6? Please proofread the manuscript carefully.
  2. Abstract: How do you correlate the quantitative value in this study to support the results? Lack of scientific support in the abstract. The authors should correlate the significant numerical results obtained in this study to support the findings mentioned in the abstract. Otherwise, it is rather a simple analysis in the Abstract.
  3. Considering legibility, the presentation of figures or graphs is lacking in the text, especially for the results.
  4. Included the limits of this research and some prospects in Section 4.

It will be a pleasure to read a revised version of the paper, in which, the above issues have been carefully addressed.

Author Response

Comments and Suggestions for Authors

The study reveals the effect of financial inclusion on environmental degradation, in the case of South Asia during 2000-2019. However, the paper must address some problems before it could be published

  1. The authors shall provide the novel contribution and significance of the study in the Introduction, not just merely highlighting ASEAN's condition. Otherwise, this would be difficult to highlight the research gaps from an international perspective and link them to the objectives of the study.

Authors’ response: Thanks for your in-depth review. The significance has been improved and the novelty has been added in the introduction section.

  1. It is suggested to give the definition or meaning of financial inclusion. ALSO, please explain the difference between financial development and financial inclusion, so readers can better understand the contribution of this study.

Authors’ response: Thank you for your comments. The difference between financial development and financial inclusion was incorporated in the revised version of the manuscript. These changes can be verified in the revised version.

  1. Why you should use ARDL-PMC estimation not other techniques to study the effects? And what are their advantages over traditional econometric methods? More explanation is necessary.

Authors’ response: Thanks for your valuable comments. Why we used the PMG-ARDL approach and what are its advantages over traditional econometric models were explained in the methodology section. These changes can be verified in the revised version of the manuscript.

  1. I am confused about the research sample for this article, how many countries did you use, all Asian countries or South Asian countries, and which countries?

Authors’ response: Thanks for your in-depth review. The sample for this article is the ASEAN region covering nine countries and excluding Laos due to lack of data on numerous variables.

  1. What is the meaning of ATM?

Authors’ response: ATM means automated teller machine. A machine that dispenses cash or performs other banking services when an account holder inserts a bank card. It is used in different parts of the world as automated bank machines (ABM) or cash machines. Automated teller machines (ATMs) are electronic banking outlets that allow people to complete transactions without going into a branch of their bank. We have made the necessary changes in the text.

  1. Move Table 3 to Section 3.

Authors’ response: Thanks for your suggestion. We have moved Table 3 to Section 3.

  1. Lack of theoretical basis in Eq. (3). According to the STIRPAT model, environmental degradation/carbon emissions will be affected by three factors: population, affluence, and technology. You can find evidence in a lot of literature. You should kindly explain why you choose these variables?

Authors’ response:  Thank you for your suggestions. Justifications for including variables that affect environmental degradation in this study were provided in the second and third paragraphs of section 3. These changes can be verified in the revised version of the manuscript.

  1. To broaden the view of readers and display the innovation in this paper, more recently published pollution emissions & environmental policy-related studies should be added and discussed, e.g., the following literature:

https://doi.org/10.1016/j.ecolind.2021.107795

https://doi.org/10.1016/j.enpol.2020.112017

https://doi.org/10.1016/j.energy.2021.121295

https://doi.org/10.1016/j.enpol.2021.112668 

The authors need to provide a structured synthesis of them in the literature review or other related parts.

Authors’ response: Thank you for your suggestions. We have incorporated suggested literature into the Introduction and Literature Review sections. These can be verified in the revised version of the manuscript.

  1. Check the Table numbers, there are two Table 6. In addition, Tables 5 and 6 should be three-line tables. What is the significance level in Tables 5 and 6? Please proofread the manuscript carefully.

Authors’ response: Thanks for your in-depth review. We corrected the table numbers and made other necessary changes. We also inserted the significance level for Table 5 and removed the * and ** significance levels for Table 6, as the probability value is greater than 5%. These changes can be verified in the revised version of the manuscript.

  1. Abstract: How do you correlate the quantitative value in this study to support the results? Lack of scientific support in the abstract. The authors should correlate the significant numerical results obtained in this study to support the findings mentioned in the abstract. Otherwise, it is rather a simple analysis in the Abstract.

Authors’ response: Thank you for your suggestions. We have revised the abstract considering the reviewer's comments.

  1. Considering legibility, the presentation of figures or graphs is lacking in the text, especially for the results.

Authors’ response: Thanks for your suggestions. We have included figures in the results and discussion section. These can be verified in the revised version of the manuscript.

  1. Included the limits of this research and some prospects in Section 4.

Authors’ response: Thank you for your suggestions. We have included limitations of this study and some prospects in Section 4. These changes can be verified in the revised version of the manuscript.

Reviewer 2 Report

Strengths:

  • The paper addresses the link between financial inclusion and environmental degradation in a rapidly developing part of the world. The work is quite important.
  • Literature review and introduction is quite well set up.

Weaknesses:

  • The major weakness is in the study design and the construction of the Financial Inclusion Index.
    • Financial Inclusion index has to deal with both individuals and businesses. The index introduced in the paper only uses personal attributes. Factors like number of business loans, amount of business loans, etc. that deal with business' financial inclusion is missing.
    • The financial inclusion index (FI) looks like a composite indicator. There needs to be a certain rigor when constructing a composite indicator. The factors that go into the indicator must be supported by theory and there are general best practices. Here is an example (although not directly applicable): https://www.oecd.org/els/soc/handbookonconstructingcompositeindicatorsmethodologyanduserguide.htm
  • Other things to note:
    • The paper deals with ASEAN countries meaning South East Asian association, but the title reads South Asia. The authors might change the title to read ASEAN countries or region.
    • ASEAN is not expanded in the paper at all. A neutral reader from outside Asia or someone not familiar with that Association would find it difficult to understand the term.
    • The introduction seems to repeat the same idea in different ways in multiple paragraphs. Please think about condensing it.
  • Grammar/English/Style Issues:
    • At the beginning of the introduction section the references sometimes have a beginning '(' but no ending ')'. The square brackets '[ ]' are already there. So, no need to include a '(' or ')'.
    • On page 2, "Based on the 2017 Global Financial Inclusion Index Report..... does not have a citation. Also, in the 2017/2018 report there is no Financial Inclusion Index report. If it is part of a sub report, the authors must clarify this.
    • ARDL-PMG and PMG-ARDL are used interchangeably without consistency. The abbreviation is not expanded in the paper.

Author Response

Comments and Suggestions for Authors

Strengths:

The paper addresses the link between financial inclusion and environmental degradation in a rapidly developing part of the world. The work is quite important.

Literature review and introduction is quite well set up.

Weaknesses:

The major weakness is in the study design and the construction of the Financial Inclusion Index.

Financial Inclusion index has to deal with both individuals and businesses. The index introduced in the paper only uses personal attributes. Factors like number of business loans, amount of business loans, etc. that deal with business' financial inclusion is missing.

The financial inclusion index (FI) looks like a composite indicator. There needs to be a certain rigor when constructing a composite indicator. The factors that go into the indicator must be supported by theory and there are general best practices. Here is an example (although not directly applicable): https://www.oecd.org/els/soc/handbookonconstructingcompositeindicatorsmethodologyanduserguide.htm

Authors’ response: Thank you for your suggestions. Financial inclusion is a broad term that encompasses several components. In view of the previous literature, different proxies and components for the construction of the financial inclusion index have been used based on the availability of data. This study also constructed the composite index for financial inclusion based on data availability. Data for the components used in the financial inclusion index were taken from the World Bank (2021) and widely used in the literature.

Other things to note:

The paper deals with ASEAN countries meaning Southeast Asian association, but the title reads South Asia. The authors might change the title to read ASEAN countries or region.

ASEAN is not expanded in the paper at all. A neutral reader from outside Asia or someone not familiar with that Association would find it difficult to understand the term.

The introduction seems to repeat the same idea in different ways in multiple paragraphs. Please think about condensing it.

Authors’ response: Thanks for your in-depth review. The sample for this article is the ASEAN region and we have changed the title of the manuscript in accordance with the reviewer's comments. ASEAN has been expanded in the revised manuscript. Necessary changes have been incorporated in the intro section. Changes can be verified in the revised manuscript.

Grammar/English/Style Issues:

At the beginning of the introduction section the references sometimes have a beginning '(' but no ending ')'. The square brackets '[ ]' are already there. So, no need to include a '(' or ')'.

On page 2, "Based on the 2017 Global Financial Inclusion Index Report..... does not have a citation. Also, in the 2017/2018 report there is no Financial Inclusion Index report. If it is part of a sub report, the authors must clarify this.

ARDL-PMG and PMG-ARDL are used interchangeably without consistency. The abbreviation is not expanded in the paper.

Authors’ response: Thanks for your in-depth review. We have removed '(' or ')' or both from the mentioned references. We inserted the citation in the text and corrected the year of reference both in the text and in the references. This study employed the PMG-ARDL approach, and we made the necessary correction. Changes can be verified in the revised version of the manuscript.

Reviewer 3 Report

This study examines the impact of financial inclusion on environmental degradation in the ASEAN region using balanced panel data.

However, I have some major remark as follows:

1) The research gap and motivation of study are not significant in this paper, by studying the connection between financial inclusion and carbon emissions.

2) Lack of novelty and significance of study in this paper.

3) The duration of this study 2000-2019 includes different economic scenarios such as economic crisis. Therefore, the result of this study may not be able to generalize the whole impact of financial inclusion on environmental degradation. The authors should consider different economic scenario in this study.

Author Response

Comments and Suggestions for Authors

This study examines the impact of financial inclusion on environmental degradation in the ASEAN region using balanced panel data.

However, I have some major remark as follows:

1) The research gap and motivation of study are not significant in this paper, by studying the connection between financial inclusion and carbon emissions.

Authors’ response: Thanks for your in-depth review. The research gap  and motivation of the study has been improved. These changes can be verified in the revised of the manuscript.

2) Lack of novelty and significance of study in this paper.

Authors’ response: Thanks for your comments. The significance has been improved and the novelty has been added in the introduction section.

3) The duration of this study 2000-2019 includes different economic scenarios such as economic crisis. Therefore, the result of this study may not be able to generalize the whole impact of financial inclusion on environmental degradation. The authors should consider different economic scenario in this study.

Authors’ response: Keeping in view past literature, different studies have been conducted to explore the impact of economic crisis and other factors on environmental degradation in different regions. But the rise of financial inclusion in recent years has attracted the attention of environmental economists to assess its role in environmental degradation. Therefore, this study was carried out with the aim of exploring the impact of financial inclusion on environmental degradation in the ASEAN region using balanced panel data for the period 2000–2019. This study has also taken into account other factors such as economic growth, energy use and urbanization to avoid misspecification of the econometric model. However, we have devised policy implications based on findings of the study and considering reviewer’s suggestions.

Reviewer 4 Report

Report on Sustainability Journal

Paper title: Assessing the Influence of Financial Inclusion on Environmental Degradation in South Asia through the Panel ARDL-PMG Approach

Manuscript Number: Sustainability-1697536

1.     General Comments.

This is a good paper which examine the impact of financial inclusion on environmental degradation in the ASEAN region using balanced panel data for the period 2000–2019. It conducts the CIPS panel unit root tests to test for stationarity and then used Westerlund and Edgerton's test for heterogeneity and cross-sectional dependence followed by ARDL-PMG approach to explore the long- and short-term effects of financial inclusion on environmental degradation. The main finding of the paper is that financial inclusion, energy use, economic growth and urbanization are causing environmental degradation in some of the selected member countries of the ASEAN region and noted some effects of increased CO2 emissions. However, the authors used all the methodologies which are commonly used in the literature. I suggest accepting the paper subject to minor revisions detailed below.

2.     Specific Comments for revision

1.     Abbreviations are either undefined and/or defined repeatedly – for example ASEAN, CADF, CIPS, PMG-ARDL are undefined

2.     A careful proof reading is required to make improvements in English exposition, improvement in typos, inconsistencies in reference style in the text and in the reference list.

3.     Lack of consistencies in expressing terminologies’, for example someplaces authors write PMG-ARDL and other places ARDL-PMG. Is there any reason of doing this or it’s a typos? Please see some recent work an application of ARDL

4.     Authors should employ some of the method/models used in recent studies for example one can see the review paper by Bhatti and Hung Do (2019) the applications of copula, EVT and Hung Do aet al. (2020) article for the ASEAN regions application in financial inclusion. Moreover, see PC Algorithm of Graph Theoretic Approach are being used (see. Fazal, R., et al (2022). Causality Analysis: The study of Size and Power based on riz-PC Algorithm of Graph Theoretic Approach. Technological Forecasting and Social Change180, 121691.

5.     Update literature review to 2022 – note there are only 2 papers (Liu et al & authors own paper with Zhra in the reference list for 2022, you can update by adding couple more – say for example Ghouse et al. 2022 paper entitled,  Green Energy consumption and  Inclusive Growth:  A Comprehensive Analysis of Multi-country Study, forthcoming in Frontier Energy.

6.     There are so many self-cited articles, please cut it down to important papers related to Pakistan and include others work -

References:

Do, H. Q., Bhatti, M. I., & Shahbaz, M. (2020). Is ‘oil and gas’ industry of ASEAN5 countries integrated with the US counterpart?. Applied Economics52(37), 4112-4134.

Bhatti, M. I., & Do, H. Q. (2019). Recent development in copula and its applications to the energy, forestry and environmental sciences. International journal of hydrogen energy44(36), 19453-19473.

Author Response

  1. Specific Comments for revision

  1. Abbreviations are either undefined and/or defined repeatedly – for example ASEAN, CADF, CIPS, PMG-ARDL are undefined.

Authors’ response: Thanks for your in-depth review. The necessary changes have been incorporated into the revised version of the manuscript taking into account the suggested comments.

  1. A careful proof reading is required to make improvements in English exposition, improvement in typos, inconsistencies in reference style in the text and in the reference list.

Authors’ response: Thank you for your suggestions. We have fixed the issues considering the mentioned suggestions.

  1. Lack of consistencies in expressing terminologies’, for example someplace authors write PMG-ARDL and other places ARDL-PMG. Is there any reason of doing this or it’s a typos? Please see some recent work an application of ARDL

Authors’ response: Thanks for your suggestions. The terminology is PMG-ARDL, and we have made necessary changes where necessary. These changes can be verified in the revised version of the manuscript.

  1. Authors should employ some of the method/models used in recent studies for example one can see the review paper by Bhatti and Hung Do (2019) the applications of copula, EVT and Hung Do et al. (2020) article for the ASEAN regions application in financial inclusion. Moreover, see PC Algorithm of Graph Theoretic Approachare being used (see. Fazal, R., et al (2022). Causality Analysis: The study of Size and Power based on riz-PC Algorithm of Graph Theoretic Approach. Technological Forecasting and Social Change, 180, 121691.

 Authors’ response: Thank you for your suggestions. We have benefitted from the mentioned literature and incorporated it into the revised version of the manuscript.

  1. Update literature review to 2022 – note there are only 2 papers (Liu et al & authors own paper with Zhra in the reference list for 2022, you can update by adding couple more – say for example Ghouse et al. 2022 paper entitled,  Green Energy consumption and  Inclusive Growth:  A Comprehensive Analysis of Multi-country Study, forthcoming in Frontier Energy.

Authors’ response: Thanks for your in-depth review. The literature review has been updated. Changes can be verified in the revised version of the manuscript.

  1. There are so many self-cited articles, please cut it down to important papers related to Pakistan and include others work -

Authors’ response: Thank you for your suggestions. We have incorporated the most recent relevant work into the revised version of the manuscript.

Reviewer 5 Report

The article deals with a topic well known in the literature - the relationship between financial inclusion and environmental degradation. The problem presented in the article can be related, in a sense, to the regulation adopted by the European Parliament and the Council on June 18, 2020 on the establishment of a framework to facilitate sustainable investment - a "green list" for sustainable economic activities, or taxonomy, and to the draft report on the extended taxonomy to support economic transformation. In this context, the reviewer has some doubts about the inclusion of the "Carbon emission" variable on the environmental degradation side, disregarding other environmental variables that indicate degradation and contribute to climate change, the more so as the term "environmental degradation" indicates a certain complex size. The second question concerns the "Financial Inclusion Index" variable. Why were such elements constituting this index chosen? The authors did not fully explain the nature of the other variables in the model (GDP, EU, URB). Are these the control variables?

Author Response

Comments and Suggestions for Authors

The article deals with a topic well known in the literature - the relationship between financial inclusion and environmental degradation. The problem presented in the article can be related, in a sense, to the regulation adopted by the European Parliament and the Council on June 18, 2020 on the establishment of a framework to facilitate sustainable investment - a "green list" for sustainable economic activities, or taxonomy, and to the draft report on the extended taxonomy to support economic transformation. In this context, the reviewer has some doubts about the inclusion of the "Carbon emission" variable on the environmental degradation side, disregarding other environmental variables that indicate degradation and contribute to climate change, the more so as the term "environmental degradation" indicates a certain complex size. The second question concerns the "Financial Inclusion Index" variable. Why were such elements constituting this index chosen? The authors did not fully explain the nature of the other variables in the model (GDP, EU, URB). Are these the control variables?

Authors’ response: (1). Thanks for your in-depth review and valuable suggestions. There are many factors responsible for environmental deterioration, such as solid waste, soil erosion, deforestation, water pollution and CO2 emissions. However, there is limited availability of data for other variables. Thus, this study considered CO2 emissions as a proxy for environmental degradation since carbon emissions is one of the main pollutants and responsible for about 75% of GHGs. (2). Financial inclusion means that individuals and businesses have access to useful and affordable financial products and services that meet their needs. The financial inclusion index was calculated using the principal component analysis (PCA) approach from four components. These include number of ATMs, number of bank branches, bank deposits and life insurance premium. These are the key financial inclusion/development factors and were selected based on data availability. (3). Gross domestic product (GDP), energy use (EU) and urbanization (URB) were taken as control variables to avoid model specification errors, as these are the main factors responsible for carbon emissions. Excluding these variables from the model leads to misleading hypotheses.

Round 2

Reviewer 1 Report

I have no further comment.

Author Response

(The authors gave the same response as above.)

Reviewer 2 Report

N/A

Author Response

(The authors gave the same response as above.)

Reviewer 3 Report

The authors have revised the manuscript according to the comments given. Therefore, I have no further comments.

Author Response

(The authors gave the same response as above.)
